# Self-reported adverse drug effects and associated factors among *H. pylori* infected patients on standard triple therapy: Prospective follow up study

**Endalew Gebeyehu** [1]☺*, **Desalegn Nigatu** [2]☺, **Ephrem Engidawork** [3]☺

**1** Department of Pharmacology, School of Health Sciences, College of Medicine and Health Sciences, Bahir Dar University, Bahir Dar, Ethiopia, **2** Department of Internal Medicine, School of Medicine, College of Medicine and Health Sciences, Bahir Dar University, Bahir Dar, Ethiopia, **3** Department of Pharmacology and Clinical Pharmacy, School of Pharmacy, College of Health Sciences, Addis Ababa University, Addis Ababa, Ethiopia

☺ These authors contributed equally to this work.
* endalew2008@gmail.com

**Data Availability Statement:** All relevant data are within the manuscript and its Supporting Information files.

**Funding:** The principal investigator, EG, has got finantial and material support from Bahir Dar and

## Abstract

### Background

One of the most common reasons for poor medication adherence and associated treatment failure of triple therapy is adverse drug effect (ADEs) of medications.

### Objective

Assessment of ADEs and associated factors during *H. pylori* eradication therapy.

### Method

Consented *H. pylori* positive adult outpatients on standard triple therapy (proton pump inhibitor, amoxicillin and clarithromycin) were involved in this facility based follow up study from May 2016 to April 2018 at Bahir Dar city in Ethiopia. Pre-developed questionnaire and formats were used to collect sociodemographic, medical information, and patient practice data before, during, and after therapy. Bivariate and backward stepwise multivariate logistic regression was used to analyze data. P-value < 0.05 at 95%CI was considered as significant.

### Result

A total of 421 patients were involved in the study. Almost 80% of the patients were urban residents. Mean (±SD) age and body weight of patients were 30.63 (± 10.74) years and 56.79 (± 10.17) kg, respectively. ADE was reported from 26.1% of the patients and of all the reported ADEs, more than 85% was manifested with gastrointestinal symptoms which include gastrointestinal discomfort(39.1%), nausea (13.6%), constipation(12.7%), diarrhea (12.9%) and anorexia(10%). Determinants of self-reported ADEs among patients in the present study were body mass index above 25 (AOR: 2.55; 95%CI (1.21–5.38), p = 0.014),

Addis Ababa Universsities for this work as any of graduate program students. The funder had no role in study design, data collection and analysis, decision to publish, or preparation of the manuscript.

**Competing interests:** The authors have declared that no competing interests exist.

duration of acid-pepsin disorder more than 3weeks (AOR: 3.57; 95%CI (1.63–7.81), p = 0.001), pain feeling during long interval between meals (AOR: 2.14; 95%CI (1.19–3.84), p = 0.011), and residence in urban area (AOR: 1.95; 95% CI (1.04–3.67), p = 0.038).

## Conclusion

Significant proportion of patients reported ADEs which commonly manifested with gastrointestinal symptoms. Consideration of patients' body mass index, duration of the disorder, period of the day when patients feel pain, and patients' area of residence could help to reduce ADEs experienced during *H. pylori* eradication therapy.

## Introduction

Approximately two thirds of the world's population is infected with *Helicobacter pylori (H. pylori)*, making it the most widespread infection in the world[1]. Infection with *H. pylori* is a major cause of upper gastrointestinal diseases which includes peptic ulcer disease, chronic gastritis, gastric cancer, and mucosa-associated lymphoid tissue lymphoma[2–4].

According to 1996 Maastricht I consensus guideline recommendation, worldwide accepted *H. pylori* eradication therapy is always a multidrug regimen[5–7]. Within this multidrug regimen, there are different aspects of *H. pylori* eradication regimens that differ in the duration and the composition of drugs exist in the available guidelines and one of which is the standard triple therapy that consists of a combination of two antibiotics and an acid-suppressant drug [8,9].

Eradication of *H. pylori* infection in patients receiving medications can often be difficult because of different pushing factors. One of the commonly reported determinants of eradication failure is poor patient adherence to a multidrug regimen usually due to adverse drug effects of medications. Other determinants being the chosen regimen type and antibiotic resistance. Sociodemographic variations of patients, duration of peptic ulcer, cigarette smoking, genetics, and presence of other chronic diseases are also reported to affect eradication therapy [10–12]. Effort has been made to improve efficacy and safety of *H. pylori* eradication therapy through exploring new first-line treatments, investigating antibiotic resistance rates, evolution of the use of adjunctive therapies, and patient counselling and follow-up, however failure of *H. pylori* eradication therapy is the prevailing problem in clinical practice[13–20].

As it is true in many other pharmacotherapies, adverse drug effects (ADEs) are one the most common factors that affect the quality of the *H. pylori* eradication therapy. Although ADEs during *H. pylori* eradication therapy have been described as well tolerated, the therapy may be associated with significant adverse effects usually revealed with gastrointestinal symptoms that could bring about poor adherence of patients to medications leading to eradication failure[21–23]. Factors which might increase the risk of occurrence of ADEs include; extremes of age, gender, multiple drugs, disease state, past history of ADEs or allergy, genetic, factors, large doses and other patient sociodemographic and medical variables[10–12].

Previous studies including ours showed that adverse drug effect is one of the factors that influences *H. pylori* eradication rate in patients receiving standard triple therapy with proton pump inhibitor, amoxicillin and clarithromycin[24,25]. In Ethiopia, there are no studies done on the determinant factors for the occurrence of ADEs during *H. pylori* eradication therapy with the standard triple therapy. Thus the present study was amid to assess self-reported adverse drug effects and its associated factors during *H. pylori* eradication therapy in patients taking standard triple therapy.

## Methods

### Ethical issues

The study was approved by the Institutional Review Board of College of Medicine and Health Sciences, Bahir Dar University (Reference No: BCS/171/08). Permission was sought from the health institutions after presentation of the ethical approval. Written consent was obtained from each volunteer adult outpatients fulfilling inclusion criteria (S1 Text). All the drugs used in eradication therapy were approved by Food, Medicine, Healthcare Administration and Control Authority (FMHACA) of Ethiopia and the treatment protocol is as per national General Hospital Guideline. Patients were informed about the benefits and risks of the study as well as their full right to withdraw from the study at any time in point without jeopardizing the care. Moreover, privacy and confidentiality were maintained through anonymity and restricting data access.

### Study design and setting

Facility based prospective follow up study was conducted from May 2016 to April 2018 in Bahir Dar, the capital city of Amhara Regional State, located 565 kilometers Northwest of Addis Ababa, the capital of Ethiopia. The study was conducted to assess adverse drug effects during *H. pylori* eradication therapy with standard triple therapy as part of the study that assess *H. pylori* eradication rate at two healthcare institutions namely Adinas General Hospital and Kidanemihret Higher Clinic both found in Bahir Dar city. The healthcare institutions were communicated officially through submitting letter of approval of the study protocol offered from Institutional Review Board of College of Medicine and Health Sciences at Bahir Dar University with reference No: BCS/171/08.

### Patients on standard triple therapy

Of the total 526 consented *H. pylori* positive patients, this study was conducted on 421 patients who completed follow up (Fig 1). All of them were adult outpatients (age ≥18 years) living in rural and urban settings and voluntarily agreed to give written consent. Those who were seriously sick or referred from other facilities as well as those who do not speak the local language (Amharic) were excluded from the study. Assessment of adverse drug effects experienced by patients were made following proton pump inhibitor (PPI)-based standard triple therapy with a regimen of PPI (omeprazole 40 mg or pantoprazole 40 mg, twice/day for 15 to 30 days), clarithromycin (500 mg), and amoxicillin (1000 mg), each twice/day for 10 or 14 days.

### Data collection and management

Structured questionnaire developed from the literature was used to collect data in both the recruitment and the follow up period (S1 Table). The questionnaire was developed in English and translated into the local language Amharic and then back to English. Pre-test of the questionnaire was done on 5% of the sample size in another healthcare institution in the study area to ensure whether the questionnaire was able what it was intended to capture and modification of questions were made accordingly. Patients' sociodemographic and medical information was collected during the first encounter. Data related to adverse drug effects, and also added-on homemade traditional remedies was collected during the follow up period on phone call and during their second encounter to check for eradication of *H. pylori* infection at the respective healthcare institutions (Fig 1). Both primary diagnosis as well as eradication of *H. pylori* after 4–6 weeks therapy was confirmed by a stool antigen test (S2 Table), which is recommended by both European and Japanese guidelines conducted according to the Manufacturer's

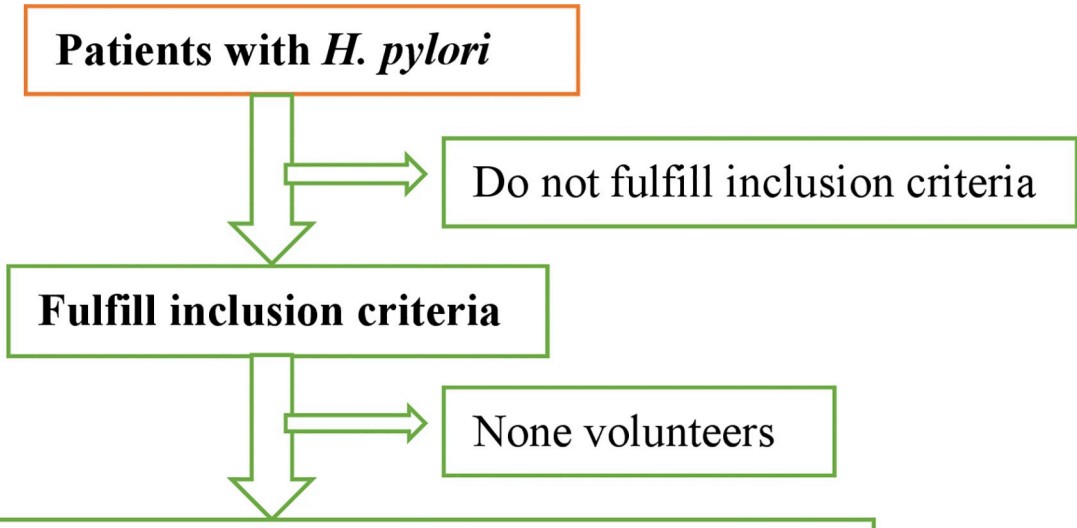

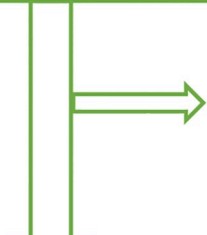

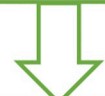

**Fig 1. Flow chart depicting sequences of the study.** HCIs: healthcare institutions.

recommendation (*SD BIOLINE H. pylori Ag*, *Standard Diagnostics*, *Inc. Korea)*[26]. Data was collected by trained clinical pharmacists and nurses. Data accuracy and consistency was assured by the study team on daily basis. Besides collecting data through completion of structured questionnaire, stool sample was collected during the second encounter of patients to determine the success or failure of *H. pylori* eradication therapy.

## Data analysis procedures

Data were entered and analyzed using SPSS statistical package version 21.0. Descriptive statistics such as percentages, means and standard deviations were used to describe data. Bivariate and multivariable logistic regressions were used to identify predictors of adverse drug effects of the standard triple-therapy. Those variables with a p-value in bivariate 0.25 were retained for multivariable logistic regression based on scientific recommendations[27]. The Hosmer-lemeshow test was checked to assess the model fattiness to conduct binary multiple logistic regression. All variables which fulfill Hosmer-Lemeshow were retained for multivariable logistic regression. Backward stepwise logistic regression model was used during multivariable logistic regression to control confounding effect. Odds ratio with 95% confidence intervals was calculated for each of the independent variables using P-value < 0.05 as the level of significance.

## Results

### Sociodemographic and medical characteristics of patients

A total of 421 patients were able to come back to the healthcare institution to confirm eradication success or failure and provide complete information related to adverse effects they had experienced during their eradication therapy. The mean age (SD) of patients was 30.63 (± 10.74) years, which ranges from 18–86. Nearly 90% of patients were under 45 years old. The mean weight of patients was 56.71 (±10.19) kg. The mean body mass index of all patients was 21.09 (±4.16).

As shown in Table 1, two-third of the patients were females and majority (80%) of the patients were urban dwellers. Close to two-third (63.4%) of them were married and a sizable proportion (42%) of them attended college education or above. Occupation wise, around 38% of the patients were employees of government and private sectors with monthly paid salary and around 62% of patients were engaged in their own income generating activities that include housewives, merchants, farmers, students, and daily laborers. Majority (86%) of the patients were followers of Ethiopian Orthodox Church.

As summarized in Table 1, almost 85% of them said that they had been living with symptoms of acid-pepsin disorder for more than 3 weeks. Almost half of the patients responded that they feel pain after meal, while 29% reported that the pain feeling persists throughout the day. About a fourth (25.6%) of the patients have responded the presence of other chronic diseases and patients response on alcohol intake before receiving triple therapy was 56.3%. One-quarter of patients reported presence of other chronic diseases of which renal impairment was the highest representing 45.4% of the cases. Almost a third (32.1%) of patients reported that they had taken diets traditionally believed to have healing effect on gastritis and peptic ulcer disease like Fenugreek and Flaxseed together with the triple therapy. Nearly two-third (66.3%) of patients received standard triple therapy for 10 days duration and the remaining for 14 days. The overall *H. pylori* eradication rate (success of the therapy) was 90% which was a bit lower than self-reported regimen completion 94.3%. However percentage of patients who reported symptom resolution of the disorder was 84.3%.

**Table 1. Sociodemographic and medical information of patients participated in the study of adverse drug effects of standard triple therapy in selected healthcare institutions at Bahir Dar City Administration, May 2016 to April 2018.** (N = 421).

| Variable and their categories | Frequency and percentage | Self-report on ADEs | | Self-reported ADEs in % |
|---|---|---|---|---|
| | | Yes (n = 110) | No (n = 311) | |
| **Sex** | | | | |
| Female | 276(65.6) | 80 | 196 | 29.0 |
| Male | 145(34.4) | 30 | 115 | 20.7 |
| **Body mass index** | | | | |
| <20 | 151(35.9) | 35 | 116 | 23.2 |
| 20–25 | 222(52.7) | 56 | 166 | 25.2 |
| >25 | 48(11.4) | 19 | 29 | 39.6 |
| **Age in years** | | | | |
| 18–24 | 125(29.7) | 33 | 92 | 26.4 |
| 25–34 | 172(40.9) | 40 | 132 | 23.3 |
| 35–44 | 75(17.8) | 23 | 52 | 30.7 |
| $\geq$ 45 | 49(11.6) | 14 | 35 | 28.6 |
| **Residence** | | | | |
| Urban | 336(79.8) | 95 | 241 | 28.3 |
| Rural | 85(20.2) | 15 | 70 | 17.5 |
| **Patents' Zonal address** | | | | |
| Bahir Dar city | 169(40.1) | 45 | 124 | 26.6 |
| West Gojjam | 96(22.8) | 22 | 74 | 22.9 |
| South Gondar | 62(14.7) | 15 | 47 | 24.2 |
| Awi zone | 52(12.4) | 16 | 36 | 30.1 |
| Others zones | 42(10.0) | 12 | 30 | 28.6 |
| **Marital status** | | | | |
| Single | 145(34.5) | 34 | 111 | 23.4 |
| Married | 267(63.4) | 73 | 194 | 27.3 |
| Divorced/Widowed | 9(2.1) | 3 | 6 | 33.3 |
| **Occupation** | | | | |
| Employee | 159 (37.8) | 46 | 113 | 28.9 |
| Non-employee | 262(62.2) | 64 | 198 | 24.4 |
| **Educational status** | | | | |
| Grade 1–8 and below | 141(33.5) | 35 | 106 | 24.8 |
| Grade 9–12 | 104(24.7) | 30 | 74 | 28.8 |
| College and above | 176(41.8) | 45 | 131 | 25.6 |
| **Time duration of the disorder** | | | | |
| $\leq$ 3weeks | 110(15.9) | 9 | 101 | 8.2 |
| >3weeks | 354(84.1) | 101 | 253 | 28.5 |
| **Presence of other disease(s)** | | | | |
| Yes | 108(25.6) | 25 | 83 | 23.1 |
| No | 313(74.3) | 85 | 228 | 27.2 |
| **Self-reported alcohol intake** | | | | |
| Yes | 237(56.3) | 52 | 185 | 21.9 |
| No | 184(43.7) | 58 | 126 | 31.5 |
| **Pain feeling period in the day** | | | | |
| After meal | 217(51.5) | 46 | 171 | 21.2 |
| Persistent in the day | 122(29.0) | 35 | 87 | 28.7 |
| Long interval b/n meals | 82(19.5) | 29 | 53 | 35.4 |

*(Continued)*

**Table 1.** (Continued)

| Variable and their categories | Frequency and percentage | Self-report on ADEs | | Self-reported ADEs in % |
|---|---|---|---|---|
| | | Yes (n = 110) | No (n = 311) | |
| **Use of Flaxseed or Fenugreek** | | | | |
| Yes | 135(32.1) | 35 | 100 | 25.9 |
| No | 286(67.9) | 75 | 211 | 26.2 |
| **Triple therapy regimen durations** | | | | |
| 10 day | 279(66.3) | 79 | 200 | 28.3 |
| 14 day | 142(33.7) | 31 | 111 | 21.8 |
| **Self-reported regimen completion** | | | | |
| Yes | 397(94.3) | 100 | 297 | 25.2 |
| No | 24(5.7) | 10 | 14 | 41.7 |
| **Disease symptom resolution** | | | | |
| Yes | 355(84.3) | 91 | 264 | 25.6 |
| No | 66(15.7) | 19 | 47 | 28.8 |

As shown in Table 1 the contribution of females and males to the reported ADEs of patients was 29.0% and 20.7% respectively. Similarly the contribution of some variable among others were; urban and rural (28.3% vs. 17.5%), disorder duration up to 3 and above three weeks (8.2% vs. 28.5%), history of alcohol intake and no intake (21.9% vs. 31.5%), 10 days and 14 days regimen (28.1% vs. 21.8%), used and not used Fenugreek or Flaxseed (25.9% vs.26.2%), symptom resolved and unresolved (25.6 vs. 28.8%) and eradication success and failure (23.7% vs. 47.6%).

As indicated in Table 2, almost a fourth (26.1%) of the patients responded that they had experienced one or more adverse drug effects while taking medications. Of the overall reported ADEs, more than 85% was gastrointestinal type which includes gastrointestinal discomfort (39.1%), nausea(13.6%), constipation(12.7%), diarrhea(12.9%) and anorexia(10%).

## Factors associated with adverse drug effects

Bivariate and multiple logistic regression analysis is shown in Table 3. On bivariate logistic regression analysis, the following variables were significantly associated with self-reported ADEs on receiving standard triple therapy: body mass index >25 (COR: 2.17 95%CI (1.01–4.33), p = 0.028); urban area residence (COR: 1.84 95%CI (1.00–3.37), p = 0.049), disease duration more than 3 weeks (COR: 2.57 95%CI (1.23–5.39), p = 0.012), history of pain feeling during long interval between meals (COR: 2.03 95%CI (1.16–3.56), p = 0.013), and history of no alcohol intake (COR: 1.24 95%CI (0.74–2.07), p = 0.027).

**Table 2. Frequency and percentage of self-reported adverse drug effects of patients received standard triple therapy in selected healthcare institutions at Bahir Dar city, May 2016 to April 2018.** (N = 421).

| Self-reported ADEs | Relative Frequency and percentage of self-reported ADEs (n = 110) | Overall percentage of self-reported ADEs (N = 421) |
|---|---|---|
| GI discomfort | 43(39.1) | 10.2 |
| Nausea | 15(13.6) | 3.6 |
| Headache and drowsiness | 15(13.6) | 3.6 |
| Constipation | 14(12.7) | 3.3 |
| Diarrhea | 12(10.9) | 2.8 |
| Anorexia | 11(10.0) | 2.6 |
| Over all | 110(100) | 26.1 |

**Table 3. Binary and multiple logistic regression analysis for factors associated with self-reported adverse drug effects on receiving standard triple therapy in selected healthcare institutions at Bahir Dar city, May 2016 to April 2018.** (N = 421).

| Variable Categories | ADEs | | Crude odds ratio* | Adjusted odds ratio** |
|---|---|---|---|---|
| | **Yes** | **No** | | |
| **Sex** | | | | |
| Female | 80 | 196 | 1.57(0.97–2.52)[g] | |
| Male | 30 | 115 | 1.00 | |
| **Body mass index** | | | | |
| <20 | 35 | 116 | 1.00 | |
| 20–25 | 56 | 166 | 1.12(0.69–1.82) | |
| >25 | 19 | 29 | 2.17(1.01–4.33)[a] | 2.55(1.21–5.34)[1] |
| **Age in years** | | | | |
| 18–24 | 33 | 92 | 0.90(0.41–1.99) | |
| 25–34 | 40 | 132 | 0.76(0.42–1.54) | |
| 35–44 | 23 | 52 | 1.11(0.50–2.43) | |
| ≥45 | 14 | 35 | 1.00 | |
| **Residence** | | | | 1.95(1.04–3.67)[2] |
| Urban | 95 | 24 | 1.84(1.00–3.37)[b] | |
| Rural | 15 | 170 | 1.00 | |
| **Zonal address** | | | | |
| Bahir Dar city | 45 | 124 | 1.00 | |
| West Gojjam | 22 | 74 | 0.82(0.46–1.47) | |
| South Gondar | 15 | 47 | 0.88(0.45–1.73) | |
| Awi zone | 16 | 36 | 1.23(0.62–2.42) | |
| Other zones | 12 | 30 | 1.10(0.52–2.34) | |
| **Marital status** | | | | |
| Single | 34 | 111 | 1.00 | |
| Married | 73 | 194 | 1.23(0.76–1.96) | |
| Divorced/Widowed | 3 | 6 | 1.63(0.39–6.90) | |
| **Occupation** | 46 | | | |
| Employee | 64 | 113 | 1.26(0.81–1.96) | |
| Non-employee | | 198 | 1.00 | |
| **Educational status** | | | | |
| Grade 1–8 and below | 35 | 106 | 0.96(0.62–1.73) | |
| Grade 9–12 | 30 | 74 | 1.18(0.49–1.46) | |
| College and above | 45 | 131 | 1.00 | |
| **Time duration of the disorder** | | | | |
| ≤ 3weeks | 9 | 101 | 1.00 | |
| >3weeks | 101 | 253 | 2.57(1.23–5.39)[c] | 3.57(1.63–7.81)[3] |
| **Pain feeling period in the day** | | | | |
| After meal | 46 | 171 | 1.00 | |
| Persistent in the day | 35 | 87 | 1.36(0.90–2.48) | |
| Long interval b/n meals | 29 | 53 | 2.03(1.16–3.56)[d] | 2.14(1.19–3.84)[4] |
| **Presence of other disease(s)** | | | | |
| Yes | 25 | 83 | 0.81(0.48–1.35) | |
| No | 85 | 228 | 1.00 | |
| **Self-reported alcohol intake** | | | | |
| Yes | 52 | 185 | 1.00 | |
| No | 58 | 126 | 1.64(1.06–2.54)[e] | |

(*Continued*)

**Table 3.** (Continued)

| Variable Categories | ADEs | | Crude odds ratio* | Adjusted odds ratio** |
|---|---|---|---|---|
| | **Yes** | **No** | | |
| **Regimen durations** | | | | |
| 10day | 79 | 200 | 1.00 | |
| 14day | 31 | 111 | 0.71(0.44–1.14)[f] | |
| **Use of Flaxseed or Fenugreek** | | | | |
| Yes | 35 | 100 | 1.00 | |
| No | 75 | 211 | 1.02(0.64–1.61) | |
| **Self-reported regimen completion** | | | | |
| Yes | 100 | 297 | 1.00 | |
| No | 10 | 14 | 2.12(0.91–4.93)[f] | |
| **Disease symptom resolution** | | | | |
| Yes | 91 | 264 | 1.00 | |
| No | 19 | 47 | 1.17(0.65–2.10) | |

*P values for binary logistic regression:

[a] = 0.028

[b] = 0.049

[c] = 0.012

[d] = 0.013

[e] = 0.027

[f] <0.25 but not significant.

**P values for multivariate logistic regression

[1] = 0.014

[2] = 0.038

[3] = 0.001

[4] = 0.011

As indicated in Table 3 above, on multivariable binary logistic regression model analysis; body mass index more than 25, duration of acid-pepsin disorder more than 3 weeks, history of pain feeling during long interval between meals, and urban area residence were significantly contributing factors for self-reported ADEs in patients on standard triple therapy. Patients with body mass index more than 25 were 2.55 (AOR: 2.55; 95%CI (1.21–5.38), p = 0.014) times more likely to report ADEs on receiving standard triple therapy compared to patients with body mass index less than 20. Patients with duration of acid-pepsin disorder more than 3weeks were 3.57 (AOR: 3.57; 95%CI (1.63–7.61), p = 0.001) times more likely to report ADEs compared with patients who stayed up to 3weeks. Patients with history of pain feeling during long interval between meals were 2.14 (AOR: 2.14; 95%CI (1.19–3.84), p = 0.011) times more likely to report ADEs compared to patients who feel pain after meal. Patients living in urban areas were 1.95 (AOR: 1.95; 95% CI (1.04–3.67), p = 0.038) times more likely to report ADEs compared to patients whose eradication therapy was successful.

## Discussion

Eradication therapy of *H. pylori* infections have proven to be difficult for different reasons. Assessment of patient and pathogen related factors could insight ways of improving treatment outcome. We evaluated self-reported ADEs that occurred during standard triple therapy and associated risk conditions of the patient that could bring about occurrence of ADEs in *H. pylori* eradication therapy. Many factors affect the occurrence of ADEs during *H. pylori*

eradication therapy regimens[28]. Understanding the different effects of these factors on ADEs enables healthcare professionals to choose most appropriate medications and give the best advice to patients[23,29].

Of 421 patients who were able to comeback for after 4–6 weeks completion of eradication therapy 110 (26.13%) reported their experience of one or more types of ADEs. These adverse effects were mild, with no documented serious adverse events. The most commonly reported ADEs were manifested with gastrointestinal symptoms which include; gastrointestinal discomfort, nausea, vomiting, diarrhea and constipation. Similar finding of gastrointestinal dominance have been reported in other studies elsewhere[23,30,31]. Self-reported ADEs in the present study was comparable 19%[32], higher than 10% -18% [33–36] and lower than 36%-76% [37–40] reported previously. The variability could be due to different factors such as duration of triple therapy, socio-demographic differences of patients, duration and severity of the disease, pharmacogenetic variability among patients, and drug combinations and possible interactions among these factors.

Predictors of self-reported ADEs of *H. pylori* eradication standard triple therapy identified in this study were female sex, urban area of residence, patient history of more than three months duration with the disease, pain feeling during long interval between meals, and residence in urban areas.

Body mass index higher than 25 was predictor of self-reported ADEs during triple therapy which could be due to more stomach distension in obese patients [41] directly linked to gastrointestinal symptoms and/or slow gastric emptying in obese patients[42] facilitating significant alteration of gastrointestinal microflora which can be responsible for reported gastrointestinal ADEs. On the other hand it could be also possible to associate higher ADEs in obese patients with slow elimination of drugs specially clarithromycin that has high tissue concentrations as reported previously ([43,44]. Negative impact of higher body mass index on *H. pylori* eradication therapy has been reported[45] which might be linked with ADEs that reduce drug intake. In addition higher *H. pylori* infection rate has been reported[1,46] in obese people. To have a better understanding of *H. pylori* infection and its eradication therapy in relation to body mass index needs further studies as it was also suggested elsewhere [47].

Patients who have been feeling pain during long interval between meals were more likely to report ADEs of medications compared with those who feel pain after meal. This could be due to reporting of disease symptoms as medication adverse effects because both conditions produce upper abdominal pain or discomfort. Similarly *H. pylori* positive patients with history of acid-pepsin disorder more than 3 weeks were more likely to report ADEs compared with those less than 3 weeks. Lower reporting of ADEs in patients with active ulcer could be relative tolerance of drugs' effects compared with more severe pain of the active ulcer and more reporting of ADEs during inactive ulcer could also be relative severity of ADEs than inactive ulcer induced pain. Although the same study is not found, study conducted on eradication of *H. pylori* in patients with active and inactive ulcers reported better rate of eradication in patients with active ulcer[48]. Parallel to this higher adherence rate has been found among the patients with acute conditions compared to those with chronic diseases[49]. Although there was no similar study that reported the effect of residence on *H. pylori* eradication with standard triple therapy some controversial reports exist in previously reported therapies[50–52]. The difference could be due to better awareness of urban patients to pay attention and reporting adverse drug effects than rural counter parts possibly associated with their differences in several aspects.

The other variable which was significant in predicting self-reported ADEs on bivariate but not on multivariate logistic regression includes history of free from self-reported alcohol intake. Patients who responded no alcohol intake prior to therapy were more likely to report

ADEs (p value = 0.027) than those reported alcohol intake. This could be due to fast elimination of drugs in patients taking alcohol because of its documented inducer effect[53,54].

There was a tendency of weak association female sex and self- reported ADEs during *H. pylori* eradication with triple therapy in this study. Although there was no similar study that reported female sex as a risk factor in development of ADEs during triple therapy, in many other therapies reviewers indicated more ADEs in females that could be due to anatomical and physiological differences such as lower bodyweight and organ size, more body fat, different gastric motility and lower glomerular filtration rate or more attentiveness to recall and report physical illness or symptom perceptions[12,55,56]. Although patients were given the same dose with the common name adult, the mean weight of females and males was 55.2(SD = 10.1) and 59.6 (SD = 9.7) respectively in this study.

There was no difference in self-reported adverse drug effects between 10 days and 14 days triple therapy regimen in this study which is similar to reported study elsewhere[10,33,57]. Following eradication therapy 84.3% of patients reported disease symptom resolution which was a bit lower than reported 91%[57]. Self-reported regimen completion was 94.3% in this study which was comparable with 95.7[58] and a bit lower than 99.8% reported elsewhere[59] and it has no significant influence on self-reported adverse effects. The response of patients showed that 84.3% of patients achieved complete resolution of acid-pepsin disorder symptoms. This value was lower than the 90% *H. pylori* eradication rate which could suggest that *H. pylori* eradication may not bring about complete symptom resolution. On the contrary acid-pepsin disorder symptoms may be resolved in patients with eradication failure.

## Limitation

Self- reported ADEs in this study could be different from the actual or reported values due to cultural and awareness differences of patients in developing and developed countries.

## Conclusion

Significant proportion of patients on standard triple therapy to eradicate *H. pylori* reported ADEs mostly manifested with gastrointestinal symptoms. Reduction of these ADEs of medications should take into account of patients' body mass index, duration of the disorder, period of the day when patients feel pain and patients' area of residence which could help to improve *H. pylori* eradication. Use of traditional homemade remedies prepared from Flaxseed or Fenugreek lacks to reduce the risk of ADEs during standard triple therapy and thus the healthcare practice shall not be influenced until this traditional practice will to be proven otherwise.

## Supporting information

**S1 File. Raw dataset.**
(SAV)

**S1 Table. Predeveloped structured questionnaire in both English and Amharic versions.**
(PDF)

**S2 Table. Format for *H. pylori* stool antigen test (SAT) data collection.**
(PDF)

**S1 Text. Written consent form.**
(PDF)

## Acknowledgments

The authors would like to acknowledge Bahir Dar University and Addis Ababa University for funding this research project. We would like to thank Adinas General Hospital and Kidane-mihret Higher Clinic for allowing data collection in their healthcare institutions. We would like to thank Abebe Fetene and Kibret Ayalew for their administrative support during data collection. Finally, we thank volunteer patients for their participation in this study project.

## Author Contributions

**Conceptualization:** Endalew Gebeyehu, Desalegn Nigatu, Ephrem Engidawork.

**Data curation:** Endalew Gebeyehu, Desalegn Nigatu, Ephrem Engidawork.

**Formal analysis:** Endalew Gebeyehu, Desalegn Nigatu, Ephrem Engidawork.

**Funding acquisition:** Endalew Gebeyehu, Ephrem Engidawork.

**Investigation:** Endalew Gebeyehu, Desalegn Nigatu, Ephrem Engidawork.

**Methodology:** Endalew Gebeyehu, Desalegn Nigatu, Ephrem Engidawork.

**Project administration:** Endalew Gebeyehu.

**Resources:** Endalew Gebeyehu, Desalegn Nigatu, Ephrem Engidawork.

**Software:** Endalew Gebeyehu.

**Supervision:** Endalew Gebeyehu, Desalegn Nigatu, Ephrem Engidawork.

**Validation:** Endalew Gebeyehu, Ephrem Engidawork.

**Visualization:** Endalew Gebeyehu.

**Writing – original draft:** Endalew Gebeyehu.

**Writing – review & editing:** Endalew Gebeyehu, Desalegn Nigatu, Ephrem Engidawork.

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
