## [Decision Letter · Decision Letter 0]

24 Oct 2019

PONE-D-19-23464

Self-reported adverse drug effects and associated factors among H. pylori infected patients on standard triple therapy: prospective follow up study

PLOS ONE

Dear Dr Gebeyehu,

Thank you for submitting your manuscript to PLOS ONE. After careful consideration, we feel that it has merit but does not fully meet PLOS ONE’s publication criteria as it currently stands. Therefore, we invite you to submit a revised version of the manuscript that addresses the points raised during the review process.

We would appreciate receiving your revised manuscript by Dec 08 2019 11:59PM. To enhance the reproducibility of your results, we recommend that if applicable you deposit your laboratory protocols in protocols.io, where a protocol can be assigned its own identifier (DOI) such that it can be cited independently in the future. For instructions see: http://journals.plos.org/plosone/s/submission-guidelines#loc-laboratory-protocols

We look forward to receiving your revised manuscript.

Kind regards,

Yan Li

Academic Editor

PLOS ONE

Journal Requirements:

4. Please include additional information regarding the survey or questionnaire used in the study and ensure that you have provided sufficient details that others could replicate the analyses. For instance, if you developed a questionnaire as part of this study and it is not under a copyright more restrictive than CC-BY, please include a copy, in both the original language and English, as Supporting Information."

Reviewers' comments:

Reviewer's Responses to Questions

**Comments to the Author**

1. Is the manuscript technically sound, and do the data support the conclusions?

Reviewer #1: No

Reviewer #2: Yes

2. Has the statistical analysis been performed appropriately and rigorously? 

Reviewer #1: I Don't Know

Reviewer #2: Yes

3. Have the authors made all data underlying the findings in their manuscript fully available?

Reviewer #1: Yes

Reviewer #2: Yes

4. Is the manuscript presented in an intelligible fashion and written in standard English?

Reviewer #1: Yes

Reviewer #2: Yes

5. Review Comments to the Author

Reviewer #1: This study aimed to know the associated factors about the adverse drug effects during Helicobacter pylori eradication. This report is interesting and can be helpful to the clinicians for increasing compliance of H. pylori eradication regimen. However, I think that some of the results and conclusions of this paper seem to be incorrect.

Comments

1. Authors analyzed the factors associated with self-reported ADE using binary and multiple logistic regression. The results showed that failure of eradication therapy was one of the associated factors for ADE and, based on the results, the authors argued that failure of eradication therapy is one of predictors of ADEs. The factors which can affect ADEs are what we can know before treatment, therefore, treatment failure can not be included. If authors want to know that ADEs can affect treatment failure, you should analyze the associated factors for success of H. pylori eradication including ADEs as variable.

2. It will be better to make flow chart as Figure 1 for easier understanding of study sequences.

Reviewer #2: In this interesting manuscript, the authors described a prospective follow up study"Self-reported adverse drug effects and associated factors among H. pylori infected patients on standard triple therapy"

the manuscript is sound and in the good writing, I don't have any further comments on this paper, recommend to be accepted .

Thank you for invitation,

6. PLOS authors have the option to publish the peer review history of their article (what does this mean?). If published, this will include your full peer review and any attached files.

Reviewer #1: No

Reviewer #2: No

---

## [Author Response · Author response to Decision Letter 0]

7 Nov 2019

We are happy on your decision of "resivion required" on our manuscript since it has given us a chance to imptove the scientific acceptance of the article. We believe that we have corrected the manuscript according to you comments. The detail of our response to reviewers is uploaded with the document file called Response to Reviewers.

---

## [Editor Report · Decision Letter 1]

8 Nov 2019

Self-reported adverse drug effects and associated factors among H. pylori infected patients on standard triple therapy: prospective follow up study

PONE-D-19-23464R1

Dear Dr. Gebeyehu,

We are pleased to inform you that your manuscript has been judged scientifically suitable for publication and will be formally accepted for publication once it complies with all outstanding technical requirements.

With kind regards,

Yan Li

Academic Editor

PLOS ONE
---

## [Editor Report · Acceptance letter]

15 Nov 2019

PONE-D-19-23464R1 

Self-reported adverse drug effects and associated factors among *H. pylori* infected patients on standard triple therapy: prospective follow up study 

Dear Dr. Gebeyehu:

I am pleased to inform you that your manuscript has been deemed suitable for publication in PLOS ONE. Congratulations! Your manuscript is now with our production department. 

With kind regards,

on behalf of

Dr. Yan Li 

Academic Editor

PLOS ONE